# A Pilot and Feasibility Study on a Mindfulness-Based Intervention Adapted for LGBTQ+ Adolescents

**DOI:** 10.3390/ijerph21101364

**Published:** 2024-10-16

**Authors:** Kasey D. Klimo, Jessica Walls Wilson, Charlotte Farewell, Rose Grace Grose, Jini E. Puma, Danielle Brittain, Lauren B. Shomaker, Kelley Quirk

**Affiliations:** 1Department of Human Development and Family Studies, Colorado State University, Fort Collins, CO 80523, USA; kasey.klimo@colostate.edu; 2Department of Community and Behavioral Health, University of Colorado Anschutz Medical Campus, Aurora, CO 80045, USA; jessica.2.wilson@cuanschutz.edu (J.W.W.); charlotte.farewell@cuanschutz.edu (C.F.); jini.puma@cuanschutz.edu (J.E.P.); 3Colorado School of Public Health, University of Northern Colorado, Greeley, CO 80639, USA; rose.grose@unco.edu; 4College of Kinesiology, University of Saskatchewan, Saskatoon, SK S7N 5A2, Canada; dani.brittain@usask.ca; 5Graduate School of Professional Psychology, University of Denver, Denver, CO 80210, USA; kelley.quirk@du.edu

**Keywords:** acceptability, feasibility, mindfulness-based interventions, LGBTQ+ health, adolescent health

## Abstract

(1) Background: Lesbian, gay, bisexual, transgender, queer and other gender and sexual minority-identified (LGBTQ+) adolescents face mental and physical health disparities compared to their heterosexual and cisgender counterparts. Mindfulness-based interventions (MBIs) may be a potential method to intervene upon health disparities in this population. This pilot study explores the initial acceptability and feasibility, along with the descriptive health changes of an online MBI, Learning to Breathe-Queer (L2B-Q), which was adapted to meet the needs of LGBTQ+ adolescents. (2) Methods: Twenty adolescents completed baseline and post-intervention assessments of mental health, stress-related health behaviors, physical stress, and LGBTQ+ identity indicators. In addition, the adolescents participated in a post-intervention focus group providing qualitative feedback regarding the acceptability of L2B-Q. (3) Results: L2B-Q demonstrated feasible recruitment and assessment retention, acceptability of content with areas for improvement in delivery processes, and safety/tolerability. From baseline to post-intervention, adolescents reported decreased depression and anxiety and improved intuitive eating, physical activity, and LGBTQ+ identity self-awareness with moderate-to-large effects. (4) Conclusions: These findings underscore the need and the benefits of adapted interventions among LGBTQ+ youth. L2B-Q warrants continued optimization and testing within the LGBTQ+ adolescent community.

## 1. Introduction

Lesbian, gay, bisexual, transgender, queer, and gender and sexuality diverse individuals (LGBTQ+) often have unique developmental experiences and risk factors related to their identities [1]. As compared to heterosexual and cisgender individuals, LGBTQ+ people often experience significant mental and physical health disparities, likely related to their experiences of discrimination, including structural and systemic barriers to mental health and medical health care [2,3]. LGBTQ+ adolescents, referring to the age span of 10–19 years of age, are at particular risk due to the often stressful nature of this dynamic during a sensitive developmental period. Stress is a normative experience for many adolescents [4]; however, the intensity, frequency and various ways in which stressors are experienced varies [5,6,7]. In addition to the normative stressors of adolescence, adolescents who identify as LGBTQ+ can face added stressors specific to their sexual and/or gender minority identity, such as dealing with societal structures built on heteronormativity [8].

Minority stress theory underscores that multiple and persistent stressors related to one’s lived experiences with a minority identity status, including discrimination, stigma and prejudice, negatively impact health [9,10]. The effects of minority stress can be especially pronounced for individuals with a sexual and gender minoritized identity [11]. A 2022 U.S. study with over 22,000 LGBTQ+ adolescents revealed that in the past year, 76% experienced verbal harassment, 54% sexual harassment (e.g., unwanted touching or sexual remarks), 31% physical harassment (e.g., pushed or shoved), and 13% physical assault (e.g., punched, kicked, or injured) based on their sexual and/or gender identity [12]. In this same cohort, 79% of LGBTQ+ students reported avoiding school functions and extracurricular activities and 43% reported missing school because they felt unsafe in their school environment [12]. These national data underscore the harrowing stressors that many LGBTQ+ adolescents in the U.S. face in their daily lives.

Such experiences of stigma, harassment, and discrimination contribute to mental health disparities within the LGBTQ+ adolescent population [9,13]. LGBTQ+ adolescents are significantly more likely to experience elevated symptoms of depression and anxiety, and worsened mental health compared to their heterosexual/cisgender peers [12,14]. In a survey of LGBTQ+ adolescents aged 14–21 years, 41% of youth had seriously considered suicide, and 14% of youth had at least one attempt in the past year [15]. Compared to heterosexual and cisgender adolescents, LGBTQ+ youth are almost five times more likely to attempt suicide [16,17]. Additionally, LGBTQ+ adolescents who experience multiple minority stressors are almost 12 times more likely to attempt suicide as compared to LGBTQ+ adolescents who do not report experiences of minority stress [18]. Unfortunately, these statistics are even more alarming for transgender and non-binary youth, with 50% of transgender youth seriously considering suicide, and one in five youth reporting a suicide attempt in the past year [15]. These data highlight an urgent need for interventions that support the mental health of LGBTQ+ youth and adolescents.

In addition, minority stressors play a role in physical health disparities. Research suggests that lesbian and bisexual women are more likely than heterosexual women to have excess weight [19]. Further, LGBTQ+ adults have higher rates of disability and cardiovascular diseases compared to heterosexual and cisgender adults [20,21,22]. There is very little research on LGBTQ+ physical health in adolescents, yet some data suggest higher rates of prediabetes and type 2 diabetes among LGBTQ+ adolescents as compared to heterosexual/cisgender adolescents [23,24]. Studies also have shown connections between the experience of greater stressors distinct to having a sexual and gender minority identity, such as family rejection and/or disapproval of one’s sexual and/or gender identity, and worse overall physical health among LGBTQ+ adolescents [25]. In addition to the social justice efforts needed to address the stressors and injustices among LGBTQ+ youth and communities at a systemic level, determining the most effective ways of supporting LGBTQ+ adolescents to cope with stressors and their effects on the body is sorely needed.

Excessive or toxic stress, referring to the extended activation of the body’s stress response system in response to chronic or overwhelming stressful events, can undermine a person’s capacity to cope effectively [26]. In the context of the toxic stress often facing people in the LGBTQ+ community, LGBTQ+ youth are more likely than cisgender youth to cope through social isolation [27]. In particular, experiences of being ‘othered,’ or being explicitly made to feel different from their peers due to their identity [28], have been related to greater social isolation [27]. This phenomenon is especially pronounced when adolescents lack an LGBTQ-inclusive community or perceive social situations as unsafe [12,27].

Additionally, some research suggests that stress-related health behaviors are heightened in LGBTQ+ individuals. Binge eating, low intuitive-eating, self-induced vomiting, and unhealthy weight control behaviors are more common among LGBTQ+ adolescents and young adults [29,30,31]. With respect to sleep, higher rates of bullying or victimization among LGBTQ+ adolescents have been related to greater disruptions in sleep patterns and lower daytime energy [32], as sleep dysregulation may be influenced by minority stress-related experiences [33]. Ultimately, both disordered eating habits and irregular sleep patterns are likely to exacerbate existing precursors for broader health disparities among LGBTQ+ youth [34], as disordered eating and poor-quality sleep have been associated with greater depression [35,36,37] and a greater risk for metabolic and coronary diseases [31,38].

The overwhelming negative health disparities experienced by the LGBTQ+ community highlight the importance of exploring how interventions can target the unique stressors LGBTQ+ people experience. In aiming to reduce stress, mindfulness-based interventions (MBIs) may be a particularly useful approach for helping LGBTQ+ adolescents navigate stressors and respond to stressors more healthfully. Rather than mitigating stressors completely in persons facing high levels of stressors, MBI delivery aims to reduce the psychological and somatic symptoms of distress and their effects on stress-related health behaviors and health outcomes [39,40,41]. MBI is focused on mindfulness training, meaning practices that promote present-moment awareness, foster a non-judgmental stance of acceptance, and allow individuals to experience thoughts and feelings in a non-judgmental way with a sense of equanimity [42,43,44]. In addition to cultivating non-judgmental, moment-to-moment mind, body, and emotional awareness, MBI targets other coping strategies such as self-compassion, healthy emotion regulation (e.g., acceptance vs. avoidance), expressing and feeling gratitude, and bodily awareness [39,40,41,44,45]. The cultivation of such strategies through MBI has led to reductions in the mental and physical health concerns of highly stressed young adults compared to a randomized control group [42].

One empirically supported MBI for adolescents is *Learning to Breathe* (L2B) [46,47,48]. L2B is centered on improving emotion regulation and attentional skills via interactive and experiential education and activities to foster mindfulness of body, thoughts, and feelings [49]. Of note, the L2B curriculum was developed for an adolescent target audience, with materials and activities geared toward this developmental period. L2B provides instruction and opportunities to learn and practice mindfulness individually (e.g., guided meditation, breath awareness) and through group activities (e.g., gentle yoga, discussion about stress and its effects). L2B has been tested in a small but growing body of studies, with data showing positive results regarding improvements in perceived stress, self-regulation skills, and mood among adolescents compared to control groups [49,50,51,52,53,54,55]. Although the preliminary results are compelling, researchers have begun to examine how adding or adapting elements of the program may strengthen these positive outcomes. In particular, adaptations that reflect the lived experiences of specific interest groups are an important ingredient for boosting the engagement and positive health effects [56]. Surprisingly, few adapted MBIs for LGBTQ+ adolescents exist [19,57]. In an effort to increase adapted MBIs for LGBTQ+ youth, and in line with intervention best practices which highlight the importance of cultural adaptation, L2B was adapted into Learning to Breathe-Queer (L2B-Q), using a term reclaimed by the LGBTQ+ community to refer to gender and sexually diverse individuals. L2B-Q was developed to target the unique stressors in the LGBTQ+ adolescent community through systematic information gathering and feedback from steering committees and focus groups of stakeholders and LGBTQ+ youth [8,58].

## 2. Materials and Methods

The primary aim of the current single-arm pilot study was to determine acceptability and feasibility of the L2B-Q program delivered to LGBTQ+ adolescents. It was predicted that L2B-Q would be acceptable to participants, as indicated by likeability and perceived utility. Safety and tolerability were monitored and reported. Also, it was expected that recruitment and retention would be feasible. The secondary aim was to describe baseline to post-intervention changes in key mental, behavioral, and identity-related indices. Although efficacy testing is beyond the scope of single-arm pilot feasibility trials [59], we hypothesized that there would be signals for clinically meaningful, positive changes in mental, behavioral, and identity-related indicators, operationalized in two ways: first, as the majority (>50%) of participants showing at least a 20% improvement from baseline to post-assessment and second, as at least small-to-moderate effect sizes (Cohen’s *d* ≥ 0.2) [60,61].

All procedures were reviewed and approved by the Colorado Multidisciplinary Institutional Review Board. Participants were recruited throughout a western area of the U.S., including rural areas (i.e., people and territories located outside a metro area) and urban areas (i.e., a metropolitan area with 50,000 or more people) [62]. Recruitment flyers and emails were sent to LGBTQ+ area organizations, including pride centers, LGBTQ+ support centers, schools, therapeutic settings serving sexual and gender minority individuals, and LGBTQ+ online community settings. Specific organizations that provided letters of support prior to beginning the study also directly reached out to eligible youth and partner organizations. Staff at partner organizations and the research team also informed potential participants about the study through word of mouth.

Interested parties who responded to recruitment efforts were contacted over the phone by members of the research team and completed screening protocols to assess eligibility. Prior to the eligibility screening, potential participants were given pre-screening consent to inform them that they could stop the eligibility inquiry at any time and that their name and identifying information would not be recorded unless they qualified for the study. Participants who were eligible to participate were middle or high school students between the ages of 12 and 18 years. They also must have identified as lesbian, gay, bisexual, transgender, gender non-conforming, queer, questioning, or another sexual or gender minority. All participants had to be residents of Colorado and needed to have access to a smart phone or computer with a webcam and stable internet access, which was necessary for L2B-Q program participation. Participants who agreed to participate in the L2B-Q intervention were provided a postcard consent document. The postcard consent document was sent electronically via a link in REDCap [63], where participants could sign the consent form electronically and download and print the consent form for their own personal records. This link also provided participants with access to the baseline survey. Given the safety concerns inherent in the LGBTQ+ adolescent population (e.g., parental rejection to identity, parental reactivity to identity, etc.) it is not uncommon to ask IRB boards to waive parental consent for treatment-based interventions [64]. Therefore, parental/guardian consent was waived for this study.

The adapted L2B-Q program was conducted with two separate cohorts of LGBTQ+ youth. The adaptation process of the L2B-Q program will be published in a different manuscript [8,58]. Both cohorts of participants engaged in 1-h long sessions conducted over secure videoconferencing for six weeks. The six sessions each had a different theme: (1) body awareness, (2) working with thoughts, (3) working with feelings, (4) awareness of thoughts, feelings, and bodily sensations, (5) tenderness and reducing harmful self-judgments, and (6) healthy habits and integrating mindful awareness into daily life. Within these themes, the L2B-Q program included the following: basic mindfulness breathing techniques to cope with stress, exploration of ruminative thoughts related to LGBTQ+ identities, identification of emotions tied to one’s identities, unpacking identity-based stress experiences, practicing coping skills, self-compassion, loving kindness as it relates to LGBTQ+ identities, and promotion of gratitude and pride around identities. Broadly, adaptations were made to each of the six modules of L2B including changing all fictional story characters portrayed in the activities to have gender neutral pronouns, and changing the stories used in the activities to portray scenarios commonly experienced by LGBTQ+ adolescents. Overall, adaptations were made to best speak to unique stressors that LGBTQ+ adolescents face, and to frame these specific stressors within contexts and characters most closely matching the experiences of sexual and gender minority adolescents.

One to seven days prior to the first session of the intervention, participants completed consent and pre-intervention questionnaires. One to seven days after the sixth session, participants completed post-intervention questionnaires. Youth also participated in a focus group one week after the final intervention session to provide insights on acceptability and to share feedback about the L2B-Q program. Focus groups were conducted over Discord, with one research assistant leading the interview, and a second research assistant transcribing participant answers. There was one week between cohort 1’s sixth session and cohort 2’s first session. All consents and surveys were completed online using REDCap software [63]. Surveys were completed at both baseline and post-intervention unless otherwise noted.

The seven-item Generalized Anxiety-7 was utilized to measure anxiety over the previous two weeks (GAD-7) [65]. Items were rated on a four-point Likert scale (1 = Not at all to 4 = Nearly every day). Higher scores indicated higher levels of anxiety, and baseline (*α* = 0.93) and post-intervention (*α* = 0.95) reliabilities were excellent.

The 8-item Patient Health Questionnaire for Depression (PHQ-8) was used to assess depression symptoms over the previous two weeks [66,67]. Items were rated on a four-point Likert scale (1 = Not at all to 4 = Nearly every day). Higher scores indicated higher levels of depression, and internal reliability was good to excellent at baseline (*α* = 0.95) and post-intervention (*α* = 0.87). In addition to baseline and post-intervention, participants completed the GAD-7 and PHQ-8 at the outset of each of the first five intervention sessions. If elevated symptoms were endorsed (GAD-7 and/or PHQ-8 score greater than 10), participants were contacted via phone by a trained member of the research team supervised by a licensed psychologist and referred to appropriate mental health resources, as indicated.

Eating behavior was assessed using the 5-item Intuitive Eating Scale [68,69]. Items were rated on a five-point Likert scale (1 = Never to 5 = Always). Lower scores indicated greater (more positive) intuitive eating. Internal reliability was good to excellent at baseline (*α* = 0.97) and post-intervention (*α* = 0.85).

Sleep was measured using the PROMIS Pediatric Sleep Disturbance Short Form 4a [70], which includes four items rated on a five-point Likert scale (1 = Never to 5 = Always). Higher scores indicated greater sleep disturbance. Cronbach’s *α* at baseline (*α* = 0.83) and post-intervention (*α* = 0.55) suggested acceptable internal consistency at baseline, but not at follow-up.

A one-item measure from the Healthy Kids Colorado Survey [71] was used to assess physical activity: “During the past seven days, on how many days were you physically active for a total of at least 60 min per day?” Answer options were between zero and seven days.

Participants completed the PROMIS Pediatric Physical Stress Experiences-Short Form 8a [72,73] to assess physical stress in the last seven days. They responded to eight items on a 5-point Likert scale (1 = Never, 5 = Always). Higher scores reflected greater physical stress experiences. Cronbach’s *α* at baseline (*α* = 0.81) and post-intervention (*α* = 0.86) suggested acceptable internal consistency.

Participants completed the Mindful Attention Awareness Scale (MAAS) [74] to assess mindful attention. The scale includes five items on a 5-point Likert scale (1 = Almost always to 5 = Very infrequently). Higher scores indicated greater mindfulness. Cronbach’s *α* at baseline (*α* = 0.77) and post-intervention (*α* = 0.92) indicated acceptable to excellent internal consistency.

Subscales of the Gender Minority Stress and Resilience Measure for Adolescents (GMSR-A) were used to examine experiences of internalized stigma and feelings of community connectedness [75]. Two subscales were used: internalized stigma (eight items) and community connectedness (five items). All items were answered on a 5-point Likert scale (1 = Strongly disagree to 5 = Strongly agree). Items on the internalized stigma subscale were adapted slightly to include gender and/or sexual identity. For example, “I resent my gender identity” was changed to “I resent my sexual and/or gender identity”. Each subscale was totaled individually, and higher scores for each subscale indicated higher levels of internalized stigma and higher levels of community connectedness. For the internalized stigma subscale, Cronbach’s *α* at baseline (*α* = 0.97) and post-intervention (*α* = 0.92) showed excellent internal consistency. For the community connectedness subscale, Cronbach’s *α* at baseline (*α* = 0.65) and post-intervention (*α* = 0.61) indicated moderate to low internal consistency.

To assess positive identity, two subscales (self-awareness and authenticity) of the Lesbian, Gay, and Bisexual Positive Identity Measure (LGB-PIM) were used [76]. A total of ten items were answered on a 5-point Likert Scale (1 = Strongly disagree to 5 = Strongly agree). Each subscale was totaled individually, and higher scores indicated a greater feeling of self-awareness and authenticity. For the self-awareness subscale, Cronbach’s at baseline (*α* = 0.95) and post-intervention (*α* = 0.96) indicated high internal consistency. For the authenticity subscale, Cronbach’s at baseline (*α* = 0.97) and post-intervention (*α* = 0.87) also indicated high internal consistency.

The focus group moderator engaged focus group participants in a discussion to gauge feasibility and acceptability based on Bowen and colleagues’ [77] framework, which includes constructs of acceptability, implementation, practicality, and demand. Specific questions were related to acceptability (i.e., Is there anything you’d like the facilitators to know about how you enjoyed the program?), implementation (i.e., What did you think of the program materials and activities shared?), and practicality (i.e., Was the time duration acceptable?). Questions about demand were not explicitly asked, but participants offered comments related to the program’s demand (i.e., participants recommending the intervention to their friends).

Descriptive statistics related to the primary quantitative outcomes, acceptability and feasibility, were analyzed for the whole sample and separately by cohort to explore any differences in acceptability between the two cohorts. Qualitative focus group data were analyzed using best practices in grounded theory and qualitative methods. Coding was an iterative process including a deductive theory-driven approach based on pre-identified feasibility constructs and a systematic inductive data-driven approach to identify patterns and themes. Specifically, themes of feasibility and acceptability (e.g., implementation, demand, practicality) were identified and coded utilizing Bowen and colleagues’ [77] guidelines on how to design a feasibility study. Transcripts of the participant responses were coded by two different coders using rapid qualitative analysis methods. In line with rapid qualitative analysis techniques, participant quotes were placed into a matrix to analyze overarching themes and to identify sub-themes [78]. The coders reviewed the quotes and their placement within the matrix as well as their interpretation as either a positive response, negative response, or a suggested change that was neutral. The coders met to discuss questions and reach consensus on any disagreements that arose from the matrices. The analyses for the secondary outcomes were based on recommendations to develop and test pilot and feasibility hypotheses without relying only on significance testing [79,80,81]. We used an approach that relies on estimation of effect sizes, estimation of confidence intervals (CIs), and calculations for changes in means from baseline to post-test. Unbiased Cohen’s *d* values along with mean differences were calculated for all variables and were used to assess statistically meaningful changes in outcomes. Due to the nature of this study, being a pilot study with a small sample size, an unbiased calculation of Cohen’s *d* is helpful in comparing outcomes to Cohen’s [82] guidelines for effect size reporting. Calculated CIs also provide information regarding the precision of the effect size estimates for all outcome variables. IBM SPSS Statistics (Version 28) was used to calculate estimated effect sizes and CIs.

## 3. Results

### 3.1. Participant Descriptives

Twenty LGBTQ+ adolescents were recruited to participate in the L2B-Q intervention and completed the baseline survey over two successive cohorts (cohort 1: *n* = 11, cohort 2: *n* = 9). All adolescents completed baseline questionnaires prior to intervention participation. At baseline, the average age was 16.5 years (*SD* = 0.94, range 14–18 years old; Table 1). The most common identities were Black (60%), male (45%), and gay (40%).

### 3.2. Primary Aim: Quantitative Indicators of Acceptability and Feasibility

CONSORT study flow is presented in Figure 1. In terms of recruitment feasibility, 27 participants were screened prior to eligibility assessment. Twenty-two participants were assessed for eligibility, with only two declining to participate in the intervention. Thus, 20 participants were enrolled within a 12-week time frame, exceeding the goal of enrolling at least ten participants within 12 weeks.

Of the 20 adolescents in the intervention phase, the median attendance was six out of six L2B-Q sessions. Eleven adolescents (55%) received 80% or more of the intervention dosage (five or six sessions), which was less than the hypothesized attendance benchmark of 80% of the sample. Eighteen adolescents completed a post-intervention assessment, reflecting a 90% assessment retention feasibility.

### 3.3. Primary Aim: Qualitative Indicators of Acceptability and Feasibility

Most participants (*n* = 16; 80%) attended a focus group discussion after session six of the intervention. Table 2 provides a summary of qualitative themes.

#### 3.3.1. Acceptability

Coders utilizing rapid qualitative analysis identified four sub-themes related to acceptability. The sub-themes of acceptability included facilitator characteristics, platform, mindful practice, and camera usage. The participants rated the facilitators highly, stating “[the facilitators] were so professional and mindful” and that they “were awesome”. Participants indicated that “Discord is awesome” and rated the platform highly. Overall, participants spoke highly of L2B-Q and the mindfulness practices. Participants indicated they “enjoyed [the program] and it was helpful”, and “it was inspirational and perhaps relieved some stress”. In contrast, participants noted a lack of comfort with the need to be on camera. For example, one participant indicated “I just wished cameras wasn’t mandated. I wasn’t comfortable”. Several other participants indicated feeling uncomfortable with cameras being encouraged during group sessions. Other than some negative responses related to camera usage, overall, the participants indicated the program was highly acceptable.

#### 3.3.2. Implementation

The sub-themes of implementation included activity changes, participant engagement, and group connection. Participants thought the activities were enjoyable, but that “some written instructions” would have been helpful prior to beginning activities. They also indicated that “having some visuals perhaps” would be helpful. When responding to questioning on how the program could be more engaging, responses included adding “songs”, and “fun activities like games”. When asked about the group connection, participants made comments that could improve the group component of the intervention, including “[the groups were] too small and sometimes we lacked proper communication”. Overall, the implementation themes suggested that the participants would like to see several changes to the program to improve activities, engagement, and group connection.

#### 3.3.3. Practicality

The sub-theme of practicality primarily related to the timing of the intervention. Participants mentioned that the timing of the intervention was “a little late” and could have been conducted “maybe an hour earlier” in the evening. Other participants indicated that they wished they could stay longer. The responses related to practicality indicated that the intervention could be further adapted to improve practicality.

#### 3.3.4. Demand

The sub-theme related to demand was peer recruitment (i.e., peer interest in L2B-Q participation). Several participants indicated that they knew of others who would enjoy the program or friends who were “mad [they] didn’t get invited [to the program]”. The responses related to demand indicated that the demand for the L2B-Q program was high.

### 3.4. Safety and Tolerability

At baseline, 9 of 20 participants (45% of the enrolled sample) endorsed elevated anxiety and/or depression symptoms, which required a check in with a licensed psychologist. At post-intervention, only four of the original nice participants indicating elevated anxiety and/or depression continued to endorse elevated anxiety and/or depression symptoms. Two of the participants who indicated elevated anxiety and/or depression at baseline did not complete the post-assessments. Of those four participants, at post-intervention, only one participant (5.5% of total sample) indicated a clinical worsening of anxiety (e.g., moving from moderate to severe anxiety).

### 3.5. Baseline to Post-Intervention Changes in Outcomes

Little’s MCAR test indicated that data were missing at random (*X*^2^ = 41.50, *p* = 0.240); thus, given the small nature of this pilot study, listwise deletion was used. Of the 18 participants who completed the post-intervention assessments, missing data ranged from 0% to 5.55% for the variables used in this study. Table 3 summarizes the changes in means, CIs, and Cohen’s *d* values for all outcome variables.

#### 3.5.1. Mental Health: Anxiety and Depression Symptoms

Anxiety symptoms showed a statistically meaningful decrease from baseline to post-intervention with a medium-to-large effect (*d* = 0.65). In clinically significant terms, 8 out of 17 participants (47%) reported a 20% or greater reduction in anxiety symptoms, and 9 out of 17 participants (53%) had a decrease in anxiety of any magnitude (ranging from 11–100%). One participant indicated a clinical worsening in anxiety symptoms. There was a statistically meaningful decrease in depression symptoms from baseline to post-intervention, which represented a large effect (*d* = 0.82). A total of 11 out of 18 adolescents (61%) reported at least a 20% reduction in depression symptoms, and 13 out of 18 (72%) reported a reduction of any magnitude (ranging from 12 to 100%). None of the participants exhibited a clinical worsening of depression symptoms.

#### 3.5.2. Mindfulness

Mindfulness had a statistically meaningful improvement from baseline to post-intervention with a small effect (*d* = −0.23). A total of 7 out of 17 participants (41%) reported a 20% or greater improvement in mindfulness. Additionally, 8 of 17 participants (47%) reported improvement in mindfulness of any magnitude (ranging from 8 to 329%).

#### 3.5.3. Stress-Related Health Behaviors: Eating, Sleep, Physical Activity

There was a statistically meaningful improvement in intuitive eating from baseline to post-intervention with a medium effect (*d* = 0.47). In clinically meaningful terms, 8 of the 18 participants (44%) indicated at least 20% improvement in intuitive eating, and 11 out of 18 (61%) showed any magnitude of improvement (ranging from 7 to 75%). Also, the adolescents experienced a statistically meaningful reduction in sleep problems from baseline to post-intervention with a small effect (*d* = 0.33). A total of 5 out of 17 participants (29%) indicated a 20% or greater improvement in sleep problems, and 8 out of 17 participants (47%) indicated an improvement in sleep problems of any magnitude (ranging from 9 to 67%). The reported physical activity frequency showed a statistically meaningful increase from baseline to post-intervention with a medium effect (*d* = 0.56). A total of 8 out of 18 participants (44%) reported a 20% or greater improvement in physical activity, and 9 out of 18 (50%) had an improvement of any magnitude (ranging from 17 to 67%). Two participants (11%) had no room for improvement, as they reported a baseline physical activity frequency of seven days a week at both baseline and post-intervention.

#### 3.5.4. Physical Stress Experiences

The participants experienced a statistically meaningful reduction in physical stress experiences from baseline to post-intervention with a small effect (*d* = 0.35). A total of 5 out of 17 adolescents (29%) reported a 20% or greater improvement in physical stress. Additionally, 11 out of 17 (65%) reported a reduction in physical stress of any magnitude (5–48%).

#### 3.5.5. LGBTQ+ Identity Variables

The adolescents’ reports of internalized shame, on average, did not change (*d* = 0.11) from baseline to post-intervention. A total of 4 out of 17 participants (14%) reported a 20% or greater reduction in internalized shame, and 8 out of 17 (47%) reported any improvement (range of 5–36% reduction). At post-intervention, the adolescents’ LGBTQ+ community connectedness scores indicated a statistically meaningful increase from baseline to post-intervention with a medium effect (*d* = 0.43). A total of 3 out of 18 participants (17%) reported a 20% or greater increase in community connection, and 10 out of 18 participants (56%) reported any improvement (ranging from 5–64%) in community connectedness. There was a statistically meaningful increase in LGBTQ+ identity self-awareness from baseline to post-intervention with a medium effect (*d* = 0.53). In clinically meaningful terms, 4 out of 18 participants (22%) reported a 20% or greater increase in self-awareness, and 9 out of 18 (50%) had any improvement (ranging from 4 to 80%). There was not a statistically meaningful change in LGBTQ+ identity authenticity (*d* = 0.17) from baseline to post-intervention. Only 2 of the 17 participants (12%) reported a 20% or greater increase in authenticity, and 6 out of 17 (35%) reported any improvement (ranging from 5 to 80%).

## 4. Discussion

The current study was a single-arm pilot and feasibility trial of a group-based mindfulness intervention adapted for LGBTQ+ adolescents: Learning to Breathe-Queer (L2B-Q), which was intended to increase mindful attention and awareness to ameliorate anxiety/depression and stress-related health behaviors. The primary outcomes were feasibility and acceptability indicators, and the secondary outcomes were descriptions of statistical and clinically meaningful changes in mental health, stress-related health behaviors, and LGBTQ+ identity-related characteristics.

The results indicated that both timely recruitment and retention for post-intervention follow-up assessments were feasible. The recruitment and enrollment rates were double the anticipated feasibility benchmark of at least 10 adolescents enrolled within a 12-week period. The study team enrolled 20 adolescents in this timeframe to two successive L2B-Q cohorts (i.e., 11 participants in cohort one, and 9 participants in cohort two). Moreover, 91% of those who were determined to be eligible by screening enrolled in the study (i.e., 20 out of 22 youth). Ninety percent of the adolescents were retained for the post-intervention assessment, with little missing data (<5%) for those who participated in the follow-up assessments. These quantitative metrics of feasibility support the timely recruitment of LGBTQ+ adolescents to a virtual L2B-Q program with feasible protocol retention to post-intervention follow-up. In the future, it will be important to determine recruitment feasibility to a randomized, controlled trial, where adolescents could receive L2B-Q versus a control or comparison condition, and the feasibility of lengthier follow-up.

In contrast, attendance to the L2B-Q sessions was lower than expected. Although the median was six out of six sessions, only 55% of participants received 80–100% of L2B-Q sessions. The qualitative themes from the focus groups are useful in interpreting this metric in context. On one hand, many aspects of L2B-Q were described with high liking and acceptability by focus group participants. No adolescent indicated a dislike for the content of the program, and they described the mindfulness components, facilitators, and the platform for the group-based virtual delivery as acceptable. Another theme was that the demand for a program like L2B-Q was perceived as high. Several participants described that they had friends who were upset they could not participate, and others said that they would tell their friends about their experience. Oppositely, an aspect of L2B-Q that was not well accepted was the encouragement to be on camera. Some participants expressed discomfort on camera or feeling shy. This feedback is consistent with the reality that the concealment of identity often serves as a way for LGBTQ+ individuals to feel safe navigating their daily lives [83,84]. It is possible that attendance was impacted by the encouragement of camera usage due to it eliciting unpleasant feelings among some youth. In future iterations of L2B-Q or virtually delivered interventions for LGBTQ+ adolescents, it will be important to implement strategies to create an affirming space for participants [85]. While camera usage may not be critical to therapeutic benefit, it does facilitate more social engagement. Indeed, implementation themes from the focus group discussions indicated that participants desired more opportunities to connect as a group, underscoring the importance of facilitating social connectedness. Interestingly, despite the lower than anticipated attendance, some participants indicated that they would have liked longer sessions. Incorporating this input on implementation will be helpful in optimizing L2B-Q delivery in the future.

Safety and tolerability were monitored throughout the intervention. Between baseline and post-intervention, fewer check-ins with a psychologist were required week-to-week. However, of the nine participants who needed a check-in at baseline, four still required check-ins at post-intervention. Only one participant (5.5% of sample) indicated a slight elevation in mental health symptoms. These findings indicate that for the majority of participants, the safety and tolerability of the intervention was observed. Although one participant experienced a slight elevation in mental health symptoms, this increase cannot be attributed to the program due to a lack of control group. However, it is important to note that L2B-Q alone may not be enough for some participants to reduce mental health symptoms, and some youth may require additional resources.

It was hypothesized that the outcome variables would improve with small to medium effect sizes and that at least half of the participants would indicate a 20% improvement in the outcomes from baseline to post-intervention. The findings varied by outcome; however, most outcome variables saw statistically meaningful and promising results.

In terms of mental health, both anxiety and depression symptoms decreased from baseline to post-intervention with large effects. Likewise, about half of the adolescents had a 20% or greater reduction in anxiety symptoms, and over 50% showed similar reductions in depression symptoms. The effects on depression and anxiety seen among the L2B-Q participants are in line with other pilot findings on the effects of L2B among at-risk youth, not identified as sexual and gender minorities, which have found that L2B demonstrated moderate-to-large effects in reducing depression symptoms [50]. Of note, a meta-analysis of MBIs for adolescents showed only small effects for reducing anxiety and depression [86]. Therefore, the results seen among the L2B-Q participants seem to confirm the mindfulness stress-buffering hypothesis [87], further indicating that MBIs may be most beneficial in populations with greater amounts of stress, including LGBTQ+ adolescents.

Interestingly, mindfulness itself among the L2B-Q participants had only a small effect/increase. Still, the findings from L2B-Q are in line with other reviews of MBIs, indicating that the effects on mindfulness are often small [60,61]. Given the discrepancy in the effects seen in depression and anxiety, other mechanisms of action (e.g., emotion regulation) should be explored in future research.

Changes in sleep disturbance, intuitive eating, physical activity, and physical stress were explored. At post-intervention, the L2B-Q participants reported improved sleep and physical stress with small effects. Intuitive eating and physical activity improved with medium effects, however less than half of the sample showed a 20% improvement in each of the aforementioned variables. These findings are in line with the literature, which indicates that MBIs may be associated with small improvements in sleep [55] and small improvements in body awareness and movement [39,40,41,44,45]. Furthermore, consistent with the implementation of MBIs in other adolescent and adult samples, this pattern indicates that MBIs may act as a way to intervene upon maladaptive eating behaviors [88,89,90,91,92] and reduce the physical effects of stress [39,40,41,42]. L2B-Q does not have the power to remove stressful experiences from the lives of LGBTQ+ adolescents; however, L2B-Q may have provided the participants with the tools to be more mindful as a way to improve sleep, intuitive eating, physical activity, and reduce the physical effects of stress. This result highlights that MBIs may be beneficial in combating stress-related health behaviors and physical manifestations of stress among LGBTQ+ youth.

Changes in LGBTQ+ identity-related variables in response to L2B-Q showed mixed results. On one hand, the adolescents reported increased feelings of connectedness to the LGBTQ+ community with small effects and increased LGBTQ+ identity self-awareness following their participation in the L2B-Q program with medium effects. Less than 50% of the sample had a 20% or greater improvement. Oppositely, there was a small increase in LGBTQ+ identity authenticity and internalized shame that was not statistically meaningful. Although some identity factors increased, these results seem to contradict previous literature, which has found associations between MBIs and increased self-acceptance and decreased shame [48,93]. Although cultural adaptation likely increases the ability of participants to interact with other similarly identifying individuals, this is unlikely to change the perception of one’s own identity in such a short time. It may be that one feels more connected and finds ways to express one’s identity with others, while also continuing to struggle to achieve real and sustained self-acceptance. However, these results highlight the important secondary impact on social connectedness that a culturally adapted program may have, beyond the intended influence on outcome variables related to the intervention curriculum. L2B-Q likely requires augmentation to support more robust, clinically meaningful effects on community connectedness and to further explore identity-related variables. Combined with the feedback from the focus groups, which indicated a greater desire for connection among participants, L2B-Q has room to enhance the facilitation of identity-related variables.

Overall, the L2B-Q participants reported improvements in both mental health and health behaviors, in addition to community connectedness, at the end of the L2B-Q intervention. These results highlight the importance of culturally adapted interventions for minoritized populations. The overall improvements in mental health and health behaviors among the LGBTQ+ adolescents from introducing mindfulness as a coping strategy are similar to the improvements seen in heterosexual adolescents. Thus, it is imperative that interventions targeting health outcomes in the LGBTQ+ population consider tailoring interventions to the specific minority stressors and experiences of the participants.

Several important limitations to this study should be noted. First, when measuring outcomes, the research relied on self-reported health behaviors, as opposed to objective measures like actigraphy, test meals, or food diaries or dietary recalls. Likewise, there were no objective measures of physical health (i.e., blood pressure, glucose regulation, lipids), which may provide added value in a future study. Self-reported mental health and health behavior data, given it represents the participants’ self-assessment of their health, is a strong proxy [94]. In addition, the research team did not collect data on how adolescents explicitly coped with or responded to stress (i.e., how they handled stressors, specific coping strategies, etc.). Therefore, the research team was unable to assess if specific coping strategies and responses to stress changed after the completion of the intervention. This single-arm pilot study, by design, lacked a control group. Future iterations of L2B-Q should utilize a control group and randomization to show stronger evidence of L2B-Q’s effects on the observed changes in outcome variables in addition to including more comprehensive assessments of both health behaviors and health effects.

Longer term follow-up to characterize the durability of the effects would also be important. The research team did not conduct a long-term follow up after the conclusion of L2B-Q. A six-month and/or one-year follow-up is needed in the future to see if the effects on mental health and health behaviors continue. Follow-ups would also help to determine if identity-related variables yield more meaningful results, or if modifying the program to address identity-related topics specifically offers different results. As is often the case, this pilot study had a small sample size. Additionally, the sample was largely urban. This characteristic is problematic due to the limited variability of influences from more diverse groups (i.e., people from more rural areas, more gender and sexually diverse individuals, and more socioeconomic diversity). The next phases of testing the efficacy and effectiveness of L2B-Q should include larger sample sizes.

## 5. Conclusions

LGBTQ+ adolescents face a variety of unique stressors in addition to typical developmental challenges. Quality evidence-based interventions that target these stressors are direly needed to prevent the subsequent health disparities observed in the LGBTQ+ community later in life. The findings presented in this study underscore the need for culturally adapted interventions to be further tested and implemented among LGBTQ+ youth. Therefore, it is imperative that L2B-Q continues to be tested and replicated within the LGBTQ+ adolescent community to further gain insights into the effects of this ground-breaking program. If these important and meaningful effects continue to be found, L2B-Q could be incredibly helpful to a highly stressed LBGTQ+ adolescent population. Further pilot testing of L2B-Q, utilizing randomized-controlled methods among rural LGBTQ+ adolescents and among LGBTQ+ adolescents living in geopolitically conservative areas where stressors may be more intense and frequent, is a necessary step in helping to mitigate health disparities among this population.

## Figures and Tables

**Figure 1 ijerph-21-01364-f001:**
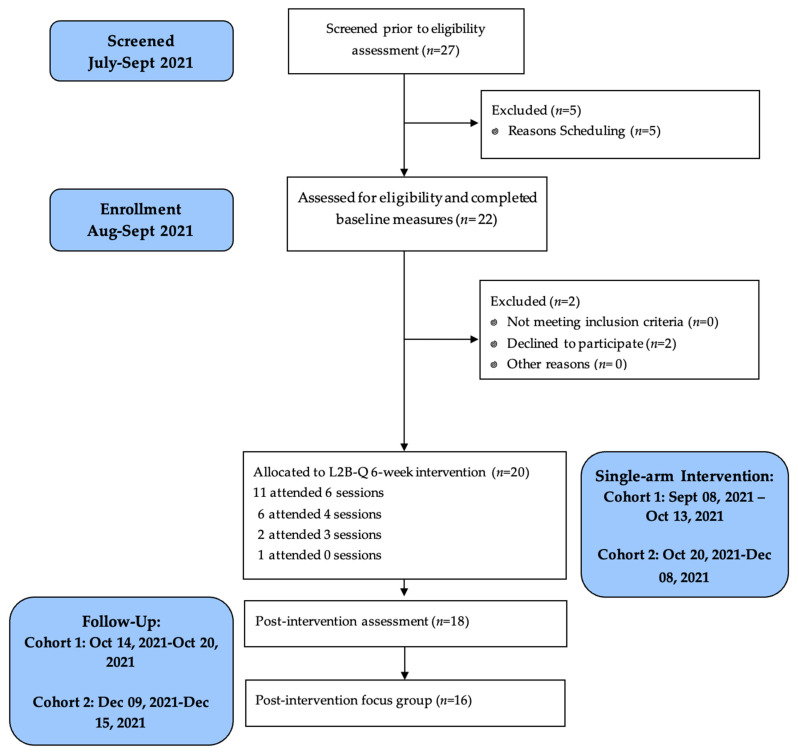
CONSORT pilot study flow of L2B-Q.

**Table 1 ijerph-21-01364-t001:** Baseline characteristics of participants.

Identity	Number (Total *N* = 20)	Percentage (%)
Gender		
Female/Woman	7	35
Male/Man	9	45
Non-Binary	3	15
Transgender	1	5
Sexual Orientation		
Lesbian	7	35
Gay	8	40
Bisexual	3	15
Asexual	1	5
Demisexual	1	5
Race		
Black	12	6
Asian	1	5
Native Hawaiian	1	5
White	6	3
Ethnicity		
Hispanic	5	2
Non-Hispanic	14	7
No Response	1	5
Age		
14	1	5
16	9	45
17	7	35
18	3	15

**Table 2 ijerph-21-01364-t002:** Focus group themes and sub-themes.

Themes	Sub-Themes	Example Responses:
Acceptability	Facilitator Characteristics	*“Looking forward to meeting every week thanks to the moderators we had”.*
		*“It was all good and the moderators were the best”.*
	Platform	*“Discord for sure”.*
		*“Discord is perfect”*
	Mindful Practice	*“These [activities] helped a lot always looking forward”.*
		*“Gratitude practice helps you be more thankful and it was a boost to my self- esteem”.*
	Camera Usage	*“Trying to stay anonymous”*
		*“I’m an introvert I don’t like being seen”.*
Implementation	Activity Changes	*“Some visuals would be nice”*
		*“Maybe some written instructions”.*
	Participant Engagement	*“Good contents and activities will always bring anyone back. It’ll keep us coming for more”.*
		*“Sometimes I get carried away by other stuff like a date night or friends’ night out”.*
	Group Connection	*“[The groups were] too small and sometimes we lacked proper communication”.*
		*“More introductions and maybe got to know each other a bit more I would be more willing to show my face”.*
Practicality	Timing	*“The time is a little late”.*
		*“I want to stay longer”.*
Demand	Peer Recruitment	*“[I will] play by my friends”.*
		*“[My friend] moved out of Colorado, she would have loved to join”.*

*Notes*: Responses include quotes from both cohorts of the L2B-Q program.

**Table 3 ijerph-21-01364-t003:** Changes in LGBTQ+ adolescents’ mental health, health behavior, and identity characteristics from baseline to post-intervention.

Variable	Baseline		Post-Intervention		*Mdiff*	Cohen’s *d*	95% CI
	M	*SD*	*M*	*SD*			
Depression	8.67	7.24	5.11	4.89	−3.56	−0.82	(−1.34, −0.27)
Anxiety	7.59	5.30	4.82	5.98	−2.76	−0.65	(−1.17, −0.12)
Mindfulness	3.40	1.20	3.78	1.31	0.38	0.23	(−0.25, 0.71)
Sleep Disturbance	10.59	2.06	9.53	3.56	−1.06	−0.33	(−0.81, 0.17)
Non-Intuitive Eating	16.39	5.71	13.33	4.93	−3.06	−0.47	(−0.03, 0.95)
Physical Activity	4.56	1.79	5.17	1.69	0.61	0.56	(0.05, 1.05)
Physical Stress	19.39	6.07	17.61	6.03	−1.78	−0.35	(−0.82, 0.14)
Internalized Shame	18.18	6.70	19.00	8.46	0.82	0.11	(−0.37, 0.58)
Community Connectedness	18.33	3.03	19.44	2.77	1.11	0.36	(−0.12, 0.83)
Positive Identity: Self-Awareness	19.00	5.20	21.72	3.46	2.72	0.53	(0.03, 1.02)
Positive Identity: Authenticity	19.76	4.80	20.71	3.12	0.95	0.17	(−0.31, 0.65)

*Note: Mdiff* is the difference in means from baseline to post-intervention, with the former subtracted from the latter. CI = Confidence interval.

## Data Availability

The raw data supporting the conclusions of this article will be made available by the authors upon request.

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
