# Peer review of "A Pilot and Feasibility Study on a Mindfulness-Based Intervention Adapted for LGBTQ+ Adolescents"

_ijerph, 2024, doi:10.3390/ijerph21101364_

Round 1
Reviewer 1 Report
Comments and Suggestions for Authors
Thank you for the opportunity to review this manuscript. I feel that it would be a valuable contribution. The only areas that I see for improvement include:
1. More updated references are needed, as many of the references are more than 5-7 years old.
2. It's unclear when the pre-test and post-test was administered. How much time elapsed before the first session was administered and before the week six deadline? Also, it's not clear how many people were in each group and at which times. Essentially, establishing a time frame is helpful.
3. I was interested to read that the authors did not need parental consent. What safeguards were put into place in case participants experienced some level of stress? Have other studies also had the parental consent process waived for this population? More information for this decision is needed.
Author Response
Response to Reviewer 1’s Comments
|
||
1. Summary |
|
|
Thank you very much for taking the time to review this manuscript. The research team appreciates your feedback and did our best to address your comments. Please find the detailed responses below and the corresponding revisions/corrections highlighted with red text in the re-submitted files.
|
||
2. Questions for General Evaluation |
Reviewer’s Evaluation |
|
Does the introduction provide sufficient background and include all relevant references? |
Yes |
|
Are all the cited references relevant to the research? |
Yes |
|
Is the research design appropriate? |
Can be improved |
|
Are the methods adequately described? |
Can be improved |
|
Are the results clearly presented? |
Yes |
|
Are the conclusions supported by the results? |
Yes |
|
3. Point-by-point response to Comments and Suggestions for Authors |
||
Comment 1: [More updated references are needed, as many of the references are more than 5-7 years old.]
|
||
Response 1: Thank you for pointing out that some of the references could be updated. The research team agreed with your comment. Therefore, we have updated several of our references to reflect the most up to date research and systematic reviews of the literature. Specifically, we have updated references 2, 3, 11, 14, 19, 24, 25, 30-32, 36, 37, 39, 41-43, 45, 46, 83, and 91. The updated references can be found in red beginning on page 16. In addition, the research team decided to remove references 33 and 40 as the updated references found better represent the claims made in our literature review. Due to these changes, the reference list has been renumbered to properly reflect the changes and updates in our references.
|
||
Comment 2: [It's unclear when the pre-test and post-test was administered. How much time elapsed before the first session was administered and before the week six deadline? Also, it's not clear how many people were in each group and at which times. Essentially, establishing a time frame is helpful.] |
||
Response 2: Thank you for this comment, as we did not specifically disclose the timeframe for pre and post-test administration in the initial version of the manuscript. To address these comments, the research team added the following the manuscript on page 4-5 lines 203-210: “One to seven days prior to the first session of the intervention, participants completed consent and pre-intervention questionnaires. One to seven days after the sixth session, participants completed post-intervention questionnaires. Youth also participated in a focus group one week prior to the final intervention session. In addition, there was one week between cohort 1’s sixth session and cohort 2’s first session.” In addition, on page 6 lines, 307-309, we mention the number of participants in each cohort, “Twenty LGBTQ+ adolescents were recruited to participate in the L2B-Q intervention and completed the baseline survey over two successive cohorts (cohort 1: n=11, cohort 2: n=9).”
On page 8, the CONSORT Flow figure specifies the dates the intervention took place, however after further review, the dates are only specific to the first cohort. The figure has been updated to include the dates of each cohort in the intervention. The updated figure is included on page 8 and in this document.
|
||
Comment 3: [ I was interested to read that the authors did not need parental consent. What safeguards were put into place in case participants experienced some level of stress? Have other studies also had the parental consent process waived for this population? More information for this decision is needed. ]
Response 3: Thank you for your comment regarding consent and the lack of parental/guardian consent. Thank you for your comment regarding consent and the lack of parental/guardian consent. Given the safety concerns inherent in the LGBTQ+ adolescent population (e.g., parental rejection to identity, parental reactivity to identity, etc.) it is not uncommon to ask IRB boards to waive parental consent to treatment-based interventions. This language was changed in the manuscript on page 4 with a citation to support the consent process. The manuscript now reads as the following,” Given the safety concerns inherent in the LGBTQ+ adolescent population (e.g., parental rejection to identity, parental reactivity to identity, etc.) it is not uncommon to ask IRB boards to waive parental consent to treatment-based interventions [64]. Therefore, parental/guardian consent was waived for this study.”
Consent was obtained twice in this study, once during the eligibility screening and a second time for those deemed eligible prior to the start of data collection and intervention sessions. The process for obtaining consent during eligibility can be found on page 4, lines 164-167, “Prior to the eligibility screening, potential participants were given pre-screening consent to inform them they could stop the eligibility inquiry at any time, and that their name and identifying information would not be recorded unless they qualified for the study.”
Discussion of informed consent can also be found on page 4, lines 172-178.The research team, based on the reviewer’s comment, believed that the statement on lines 172-178 could be changed to more clearly describe the consent process, therefore, the statement was changed to the following: “Participants who agreed to participate in the L2B-Q intervention were provided a postcard consent document. The postcard consent document was sent electronically via a link in REDCap [63] where participants could sign the consent form electronically and download and print the consent form for their own personal records. This link also provided participants with access the baseline survey. Given the safety concerns inherent in the LGBTQ+ adolescent population (e.g., parental rejection to identity, parental reactivity to identity, etc.) it is not uncommon to ask IRB boards to waive parental consent to treatment-based interventions [64]. Therefore, parental/guardian consent was waived for this study”
In relation to safety and safeguards, the research team enacted safety checks each week to check on the mental wellbeing of all participants. As mentioned on page 5, lines 221-225 “ In addition to baseline and post-intervention, participants completed the GAD-7 and PHQ-8 at the outset of each of the first five intervention sessions. If elevated symptoms were endorsed (GAD-7 and/or PHQ-8 score greater than 10), participants were contacted via phone by a trained member of the research team supervised by a licensed psychologist and referred to appropriate mental health resources, as indicated.” |
||
|
Reference Added:
64.: Smith, A. U., & Schwartz, S. J. (2019). Waivers of parental consent for sexual minority youth. Accountability in Research, 26(6), 379–390.
Thank you again, Reviewer #1 for your comments, questions, and feedback. We hope that we addressed your feedback to your satisfaction, and we believe that your implemented suggestions have made our manuscript stronger.
Reviewer 2 Report
Comments and Suggestions for Authors
Thank you for the opportunity to review the submitted manuscript. The research on building inclusive strategies is crucial in addressing the significant uncertainty that youth face at various levels-personal, relational, community, systems/policy, and society/global. This work holds immense importance in guiding our understanding and interventions for the changing dynamics of youth.
Summary: The mental health (MH) challenges faced by LGBTQ+ youth, as highlighted in global literature, are a significant concern. To address this, educators, parents, and MH professionals have been utilizing Mindfulness-Based Interventions (MBI) and the L2B classroom program, which focuses on compassion and trauma-informed principles. This pilot study, involving 20 participants, uses the L2B online resource, adapted to cater to the unique needs of LGBTQ+ youth, to assess the acceptability and feasibility of the L2B-Q program. The primary goal of this research is to improve the mental health outcomes of LGBTQ+ youth, underscoring the importance of the L2B-Q program in this context.
The author's background spans the US and Canada – Departments of Human Development, Kinesiology, and Psychology. No further disclosure of the position is noted in the paper.
Minority stress theory is noted as a position informing the work as often used in the sexual minority area of study relating to the discrepancy, tensions, and conflict – noted as stressors - surface between minority values and that of the dominant culture. The stressors and impacts are physical, emotional, and social – all of which can impact an individual’s MH. The cumulative effect of stressors on the minority population impacts physical health – which is known to contribute as a protective mechanism for MH. Toxic stress can commence at any stage in ‘life’ with adverse childhood events being one further example – for this population, toxic stress impacts the second critical life-course development stage, resulting in the risk to current and future health outcomes. The management of toxic minority cumulative stressors by youth can result in less positive health behaviours – that also have poor long-term physical and mental outcomes.
MBI is an approach noted in the literature and the L2B six-week curriculum has been updated since development approx. ten years ago. Single-arm pilot, with no longer term follow up, to look at acceptability and feasibility of a modification to the program to target as specific population;
1. The L2B-Q would be acceptable (liked and want to use) to the population of study
2. Recruitment would be possible (there is a target audience and need).
3. Baseline changes (pre-post testing)
4. Hypothesis that clinical efficiency testing could be explored in depth in future studies from a position of positive change (recognised limitation in assumptions here – wanted to explore the preliminary data to determine future investigation).
Recruitment Western area US, recruitment via existing networks that support LGBTQ+. Two cohorts, measures (GAD-7; PHQ-8, eating scale, sleep PROMIS, HKCS, stress PROMIS, MAAS, subscales on GMSR-A, LGB-PIM). Overall, a promising expansion to the existing L2B program that needs further study.
Question: line 135 The authors note that there are few adaptations for LGBTQ+ but do not offer any references for the ‘few’.
Question: line 156, is this an inference that this state or states are areas in society that are less tolerant than other states, since as a non-US reviewer, does this impact L2B-Q use (is it likely to be restricted in schools in the future, as an example)?
Question: line 156 rural and urban – is there scope to give details since there are numerous definitions of these two terms and they can include very different population rates.
Question line 162: word of mouth – please explain was this in different settings or via passive recruitment posters or via people discussing at the partner organizations.
Question line: noted the participants needed phone or computer – but later notes that they needed/ could line 175 download and print the consent form – how was this returned? And how did this support participant confidentiality?
Question general: is there scope to have a table of the existing L2B and the adapted L2B-Q themes that are in the MDI approach to see easily adaptations to the existing program. Were the nine measures used in L2B-Q all different from the primary measured used in L2B-Q at its primary development and overlap (i.e.., stress, sleep, eating), noted that broad adaptations were made to the 6 themes in the program delivery (Line 189).
Question: Can I ask how these were made by an expert panel of people on the team having experience with LGBTQ+, based on the literature alone? Were these adaptations reviewed to reach a consensus in any way with recipients of L2B-Q or a subgroup of those very familiar with teaching and delivering the L2B as it currently is used (i.e., in schools)?
Question line 199: Focus groups there were 2 RA’s are they included in the authorship? What skills training did they have and how was this supported by the wider team and as we note in the emerging literature – situating ‘our’ selves and knowing the bias that we bring, is difficult to determine as we do not know about the authors positioning (i.e., profession, gender, race etc.)
Question line: 196-200 pre-and post questionnaires how soon before and after data collected? FG how long after the 6 weeks were ‘finished’.
Question line 270: two cohorts were used, sorry if I have missed it but why two and what was different about the two in some way as I am unclear as to the variation – maybe just delivery size of the L2B-Q as there is note of comparison between the two cohorts for acceptability.
Question line 273-316: Please explain and reference “best practices in grounded theory and qualitative methods” Was this deductive based on concepts of ‘feasibility and acceptability’ as GT is generative? Note ref 77 to support rapid QR. The position of interpretation is noted. I am not assured that ref 76 sufficiently supports the statement ‘best practices in grounded theory and qualitative methods’. I wonder if the matrix drawn from the work by Browne – choosing 4 of the eight areas of focus (Table 1) needs to be more explicit, since the coding framework was established and then those doing analysis were looking for evidence – are the sub-themes in Table 2 in the paper the interpretation ‘themes’ from the study? How did the team reach consensus with the themes and examples? Are all the examples in Table 2 from across both cohorts? Was there anything different between the cohorts? Is there any way to identify the participants (since all the examples could be from just one participant – which influences confidence in the data – I know it’s not likely, but I cannot tell).
Question lines 325-329: The analysis is very statement-focused rather than interpretive; for example, with the narrative that the visual was a concern for some –the interpretation could be safety and links well with the primary theories noted in the opening section. The discussion section draws on this 447-456 in that the descriptors are used, such as ‘shy’ and ‘discomfort’ with reference to concealment. While reading the paper the opportunity to see the interpretation in the results could be strengthened.
The statistical analysis I will defer to other reviews. Thee are also other programs and areas that use MDI with for example there are mindfulness centres i.e., The BC Children's Hospital Centre for Mindfulness (kelty.link) – so while there is a wide scope for various approaches to MDI and education (L2B) is a key player in this there are ways in which building wider partnerships and sharing knowledge across sectors to strengthen supports for LGBTQ+ individuals could be very useful in the future.
Author Response
Response to Reviewer 2’s Comments
|
||
1. Summary |
|
|
Thank you very much for taking the time to review this manuscript. The research team appreciates your feedback and did our best to address your comments. Please find the detailed responses below and the corresponding revisions/corrections highlighted with red text in the re-submitted files.
|
||
2. Questions for General Evaluation |
Reviewer’s Evaluation |
|
Does the introduction provide sufficient background and include all relevant references? |
Yes |
|
Are all the cited references relevant to the research? |
Yes |
|
Is the research design appropriate? |
Yes |
|
Are the methods adequately described? |
Must be improved |
|
Are the results clearly presented? |
Can be improved |
|
Are the conclusions supported by the results? |
Yes |
|
3. Point-by-point response to Comments and Suggestions for Authors |
||
Comment 1: [line 135 The authors note that there are few adaptations for LGBTQ+ but do not offer any references for the ‘few’..]
|
||
Response 1: Thank you for pointing out that we are missing citations to previous interventions that were adapted for LGBTQ+ youth. The research team apologizes for this oversight. In order to correct this error, we have revised the statement on page 3 line 135-136 to read the following: “Surprisingly, few adapted MBIs for LGBTQ+ adolescents exist [19, 57]. The citations are for two adapted programs, one focusing on mindfulness and the other promoting coping strategies, both for LGBTQ+ youth. These were the only two programs that the research team could cite that were programs specifically adapted for LGBTQ+ youth that promote mindfulness and coping. Reference added: 57. Iacono, G. Tuned In! An Affirmative Mindfulness Intervention for Sexual and Gender Diverse Young People. Journal of LGBT Youth 2024, 1-26.
|
||
Comment 2: [line 156, is this an inference that this state or states are areas in society that are less tolerant than other states, since as a non-US reviewer, does this impact L2B-Q use (is it likely to be restricted in schools in the future, as an example)?]
Response 2: Thank you for your question regarding tolerance of LGBTQ+ identity. The research team appreciates your question regarding this topic and agrees that it is a topic of importance when discussing implementation of the L2B-Q program. The difference in tolerance for LGBTQ+ individuals by area/state would potentially impact L2B-Q’s usage in schools if there is legislation that prohibits LGBTQ+ education or materials being dispersed in schools (i.e. Don’t Say Gay Bill in Florida). However, the program was adapted to be delivered online, specifically to address the issue of accessibility depending on the LGBTQ+ youth’s ability to access or find LGBTQ+-specific resources. Having the intervention online would allow youth from states/areas that are less tolerant of LGBTQ+ issues to still access the program. Therefore, the research team mentioned the general area of the U.S. in an effort to 1.) blind the review, and 2.) to provide an idea to the level of support for LGBTQ+ issues in the area the research took place.] |
||
Comment 3:[ line 156 rural and urban – is there scope to give details since there are numerous definitions of these two terms and they can include very different population rates.
Response 3: The research team appreciates your comment, and we agree that some clarification on the two terms could be added to our paper. In order to clearly define rural and urban, we have cited the definitions of the terms in the text and referred to their definitions in text to provide greater context to the terms. The updated sentence on lines 155-158 now reads, “Participants were recruited throughout a western area of the U.S., including rural areas (i.e., people, housing, and territories outside a metropolitan area) and urban areas (i.e., metropolitan area with 50,000 or more people) [62].”
Reference added: 62. Health Resources and Service Administration: Defining Rural Population. Available online: https://www.hrsa.gov/rural-health/about-us/what-is-rural#:~:text=The%20Census%20does%20not%20define,UCs)%20of%202%2C500%20%2D%2049%2C999%20people (accessed on 01 October 2024).
Comment 4: [ line 162: word of mouth – please explain was this in different settings or via passive recruitment posters or via people discussing at the partner organizations.]
Response 4: Thank you for your suggestion to further clarify the ‘word of mouth’ recruitment process. The research team agrees that this statement could be further clarified. The sentence on page 4 line 162 “In addition, potential participants were informed about the study through word of mouth,” has been changed to the following: “Staff at partner organizations and the research team also informed potential participants about the study through word of mouth."
Comment 5 [ line: noted the participants needed phone or computer – but later notes that they needed/ could line 175 download and print the consent form – how was this returned? And how did this support participant confidentiality?]
Response 5: Thank you for your questions regarding the consent process. The research team agrees that the current wording of the informed consent process added some confusion as to how it supported confidentiality. In order to address this item, we rephrased the statement about consent to further clarify how consent was obtained, and confidentiality was maintained. The statement on lines 174-175, “Participants who agreed to participate in the L2B-Q intervention were provided a postcard consent document. The postcard consent document was sent electronically via a link in REDCap [63] where participants could download and print the consent form, as well as access the baseline survey,” and was changed to the following statement, “Participants who agreed to participate in the L2B-Q intervention were provided a postcard consent document. The postcard consent document was sent electronically via a link in REDCap [63] where participants could sign the consent form electronically and download and print the consent form for their own personal records. This link also provided participants with access the baseline survey.”
Comment 6: [general: is there scope to have a table of the existing L2B and the adapted L2B-Q themes that are in the MDI approach to see easily adaptations to the existing program.]
Response 6: The research team appreciates your question regarding the specific adaptation process to create the L2B-Q program. The research team has one manuscript describing the adaptation process in detail that is being prepared for publication, and a second paper describing the results from the focus groups used in the adaptation under review. Given that, the research team believes that describing the adaptation process in detail is outside of the scope of this paper. We have blinded the references that are included related to focus group outcomes and the adaptation process. Due to the adaptation process is being detailed in another manuscript being prepared for publication by the research team, the research team edited the sentences on page 4 lines 184-185 to say the following: “The adapted L2B-Q program was conducted with two separate cohorts of LGBTQ+ youth. The adaptation process of the L2B-Q program will be published in a different manuscript [58]. The themes of the program can also be found on page 4 lines 187-195.
Comment 7: [Were the nine measures used in L2B-Q all different from the primary measured used in L2B-Q at its primary development and overlap (i.e.., stress, sleep, eating), noted that broad adaptations were made to the 6 themes in the program delivery (Line 189).]
Response 7: Thank you for your question related to the methodology of the study. The research team is happy to clarify the measurement tools utilized in the study by the participants. Broad adaptations were made to the parent program (L2B) which led to the creation of the L2B-Q program which is described on page 3 lines 136-141. A more detailed telling of the adaptation process and the stakeholders and focus group data utilized to create the adapted L2B-Q program are being published in two separate manuscripts [8, 58]. Thus, the L2B-Q was adapted prior to any participants enrolling in the intervention. Both cohorts of participants received the same L2B-Q program curriculum and completed the same sets of measurement tools at pre-and post-test. There were minor changes to the intervention between cohorts which will be detailed in the manuscript regarding the adaptation process. If there is a way this can be clarified further, the research team would be happy to oblige.
Comment 8: [ Can I ask how these were made by an expert panel of people on the team having experience with LGBTQ+, based on the literature alone? Were these adaptations reviewed to reach a consensus in any way with recipients of L2B-Q or a subgroup of those very familiar with teaching and delivering the L2B as it currently is used (i.e., in schools)?]
Response 8: Thank you for your questions regarding the adaptation process. The research team agrees that the adaption process is important to mention. Therefore, the following has been added on page 3 lines 140-142, “L2B-Q was developed to target the unique stressors in the LGBTQ+ adolescent community through systematic information gathering and feedback from steering committees and focus groups of stakeholders and LGBTQ+ youth [citations].” The research team has a manuscript in preparation detailing the specific adaptation process and the model utilized to inform the adaptation, and a second paper under review that describes the outcomes for the focus group data utilized to inform the adaptations that were used to create L2B-Q. These papers are referenced in this manuscript, in the aforementioned quote, but are blinded for review.
Comment 9: [Focus groups there were 2 RA’s are they included in the authorship? What skills training did they have and how was this supported by the wider team and as we note in the emerging literature – situating ‘our’ selves and knowing the bias that we bring, is difficult to determine as we do not know about the authors positioning (i.e., profession, gender, race etc.)]
Response 9: Thank you for your question regarding the RA’s who assisted with our study. The RA’s were not included in authorship but are mentioned in the acknowledgements as their contributions did not meet the requirements for authorship. The RAs were members of the queer community and had prior experience in quantitative and qualitative research methods and working on projects utilizing mixed-methods. The RA’s who assisted in the focus groups were trained in the L2B intervention as well as the L2B-Q intervention. The RA leading the focus group was provided a list of questions prepared by the research team to ask during the focus group, and the second RA was in charge of transcription and was trained to transcribe the information being provided by the youth without interpretation of what was being stated.
Comment 10 [line: 196-200 pre-and post questionnaires how soon before and after data collected? FG how long after the 6 weeks were ‘finished’.]
Response 10: Thank you for this comment, as we did not specifically disclose the timeframe for pre and post-test administration in the initial version of the manuscript. To address these comments, the research team added the following the manuscript on page 4-5 lines 203-210: “One to seven days prior to the first session of the intervention, participants completed consent and pre-intervention questionnaires. One to seven days after the sixth session, participants completed post-intervention questionnaires. Youth also participated in a focus group one week prior to the final intervention session...In addition, there was one week between cohort 1’s sixth session and cohort 2’s first session.”
Comment 11: [ line 270: two cohorts were used, sorry if I have missed it but why two and what was different about the two in some way as I am unclear as to the variation – maybe just delivery size of the L2B-Q as there is note of comparison between the two cohorts for acceptability.]
Response 11: Thank you for your question regarding the clarification of the two-cohort model. The research team is happy to clarify this further. Yes - small adaptations to L2B-Q were made in between iterations and will be presented/described in the subsequent paper referenced above. All primary content of the L2B-Q program and all questionnaires administered to participants remained the same.
Comment 12: [ Question line 273-316: Please explain and reference “best practices in grounded theory and qualitative methods” Was this deductive based on concepts of ‘feasibility and acceptability’ as GT is generative? Note ref 77 to support rapid QR. The position of interpretation is noted. I am not assured that ref 76 sufficiently supports the statement ‘best practices in grounded theory and qualitative methods’. I wonder if the matrix drawn from the work by Browne – choosing 4 of the eight areas of focus (Table 1) needs to be more explicit, since the coding framework was established and then those doing analysis were looking for evidence – are the sub-themes in Table 2 in the paper the interpretation ‘themes’ from the study? How did the team reach consensus with the themes and examples? Are all the examples in Table 2 from across both cohorts? Was there anything different between the cohorts? Is there any way to identify the participants (since all the examples could be from just one participant – which influences confidence in the data – I know it’s not likely, but I cannot tell).]
Response 12: Thank you for your questions regarding the qualitative data analytic process utilized in our study. The research team agrees some clarification on this process could be added to the paper. In order to add further clarification to the qualitative coding process the following was added to page 6, lines 281-285, “Qualitative focus group data were analyzed using best practices grounded in theory and qualitative methods. Coding was an iterative process including a deductive-theory driven approach based on the pre-identified feasibility constructs and a systematic inductive data-driven approach to identify patterns and themes.” Furthermore, the following was changed on page 6, lines 290-293, “The coders reviewed the quotes and their placement within the matrix as well as their interpretation as either a positive response, negative response, or a suggested change that was neutral. The coders met to discuss questions and reach consensus on any disagreements that arose from the matrices.”
To further clarify and answer the questions mentioned in the comment, example quotes in Table 2 on page 9 are from data across both cohorts and no substantial difference in themes was identified between the two cohorts. There is no way to identify the participants, but the facilitators were trained in facilitation of focus groups and elicited feedback from all participants.
Comment 13: [lines 325-329: The analysis is very statement-focused rather than interpretive; for example, with the narrative that the visual was a concern for some –the interpretation could be safety and links well with the primary theories noted in the opening section. The discussion section draws on this 447-456 in that the descriptors are used, such as ‘shy’ and ‘discomfort’ with reference to concealment. While reading the paper the opportunity to see the interpretation in the results could be strengthened.]
Response 13: Thank you for your comments regarding the analysis approach in our study. The research team wants to do our best to address your concerns. Due to extent of the mixed-methods data presented in this paper, we had to be concise in our description of the analyses and interpretation of the qualitative results. However, we agree that safety is an important theme related to this finding and note this in the discussion on page 13 lines 463-467 which have been minorly edited for grammar, “Oppositely, an aspect of L2B-Q which was not well accepted was the encouragement to be on camera. Some participants expressed discomfort on camera or feeling shy. This feedback is consistent with the reality that concealment of identity often serves as a way for LGBTQ+ individuals to feel safe navigating their daily lives [83, 84].’ In an effort to further connect to the theme of safety/discomfort, the following statement on page 13 lines 467-468 has been edited, “It is possible that attendance was impacted by the encouragement of camera usage due to eliciting unpleasant feelings among some youth.”
We have a subsequent paper under review [citation #19] that goes into depth on the qualitative findings and lived experience of queer youth in this sample based on the minority stress model that will further touch on the themes of safety and the interpretation of focus group data.
Thank you again, Reviewer #2 for your comments, questions, and feedback. We hope that we addressed your feedback to your satisfaction, and we believe that your implemented suggestions have made our manuscript stronger.
|
Reviewer 3 Report
Comments and Suggestions for Authors
Good study.
please explain and include the following in manuscript
1. What is the sample size of study.
2.The age group of participants is 12-18. those who are minors how did you obtain consent. have you obtained from the parents/ guardian permission too?
3. screened and Recruited via phone? can you please explain
4. What is your future goals? to work with more sample?
5. Lesbian/ gay/ .... among them who are having more stress? have you compared? if not can you compare and add a table?
Author Response
Response to Reviewer 3’s Comments
|
||
1. Summary |
|
|
Thank you very much for taking the time to review this manuscript. The research team appreciates your feedback and did our best to address your comments and implement changes. Please find the detailed responses below and the corresponding revisions/corrections highlighted with red text in the re-submitted files.
|
||
2. Questions for General Evaluation |
Reviewer’s Evaluation |
|
Does the introduction provide sufficient background and include all relevant references? |
Yes |
|
Are all the cited references relevant to the research? |
Yes |
|
Is the research design appropriate? |
Can be improved |
|
Are the methods adequately described? |
Can be improved |
|
Are the results clearly presented? |
Can be improved |
|
Are the conclusions supported by the results? |
Can be improved |
|
3. Point-by-point response to Comments and Suggestions for Authors |
||
Comment 1: [ What is the sample size of study.]
|
||
Response 1: Thank you for your question regarding the sample size of the study. The sample size of the study is 20 participants at baseline and 18 as post-test. The baseline number of participants can be found in Table 1 on page 7, and pre-and-posttest number of participants can be found in the CONSORT Flow Figure on page 7. In addition the sample size of the study is named in the text on page 6 lines 294-296, “Twenty LGBTQ+ adolescents were recruited to participate in the L2B-Q intervention and completed the baseline survey over two successive cohorts (cohort 1: n=11, cohort 2: n=9),” and again on page 8 lines 307-311, “Of the 20 adolescents in the intervention phase, median attendance was six out of six L2B-Q sessions. Eleven adolescents (55%) received 80% or more of the intervention dosage (five or six sessions), less than the hypothesized attendance benchmark of 80%. Eighteen adolescents completed a post-intervention assessment, reflecting 90% assessment retention feasibility.”
|
||
Comment 2: [The age group of participants is 12-18. those who are minors how did you obtain consent. have you obtained from the parents/ guardian permission too?]
Response 2: Thank you for your comment regarding consent and the lack of parental/guardian consent. Given the safety concerns inherent in the LGBTQ+ adolescent population (e.g., parental rejection to identity, parental reactivity to identity, etc.) it is not uncommon to ask IRB boards to waive parental consent to treatment-based interventions. This language was changed in the manuscript on page 4 with a citation to support the consent process. The manuscript now reads as the following,” Given the safety concerns inherent in the LGBTQ+ adolescent population (e.g., parental rejection to identity, parental reactivity to identity, etc.) it is not uncommon to ask IRB boards to waive parental consent to treatment-based interventions [64]. Therefore, parental/guardian consent was waived for this study.”
Consent was obtained twice in this study, once during the eligibility screening and a second time for those deemed eligible prior to the start of data collection and intervention sessions. The process for obtaining consent during eligibility screening can be found on page 4, lines 164-167, “Prior to the eligibility screening, potential participants were given pre-screening consent to inform them they could stop the eligibility inquiry at any time, and that their name and identifying information would not be recorded unless they qualified for the study.”
Discussion of informed consent for enrolled participants can also be found on page 4, lines 172-178. The research team, based on the reviewer’s comment, believed that the statement on lines 172-178 could be changed to more clearly describe the consent process, therefore, the statement was changed to the following: “Participants who agreed to participate in the L2B-Q intervention were provided a postcard consent document. The postcard consent document was sent electronically via a link in REDCap [63] where participants could sign the consent form electronically and download and print the consent form for their own personal records. This link also provided participants with access the baseline survey. Given the safety concerns inherent in the LGBTQ+ adolescent population (e.g., parental rejection to identity, parental reactivity to identity, etc.) it is not uncommon to ask IRB boards to waive parental consent to treatment-based interventions [64]. Therefore, parental/guardian consent was waived for this study” |
Reference added:
- Smith, A. U., & Schwartz, S. J. (2019). Waivers of parental consent for sexual minority youth. Accountability in Research, 26(6), 379–390.
Comment 3: [screened and Recruited via phone? can you please explain ]
Response 3: Thank you for your question regarding the screening and recruitment process. We agree that some additional details could be added to the recruitment steps on page X in order to further clarify the process. The sentence on page 4, lines 163-164,” Interested parties who responded to recruitment efforts were contacted over the phone and screened for eligibility,” was changed to the following: “Interested parties who responded to recruitment efforts were contacted over the phone by members of the research team, and completed screening protocols to assess eligibility.”
Comment 4: [What is your future goals? to work with more sample?]
Response 4: Thank you for your interest in our research team’s future goals for the L2B-Q program. As of now, the future goals are to further adapt the program based on feedback and outcomes found in this manuscript. Additionally, the research team has two papers that are in preparation and under review in hopes to expand this work to a larger RCT pilot test. Ideally, this larger RCT pilot study would include a larger sample of youth from a wider range of locations across the U.S. given the difference in resources, tolerance, and acceptance for LGBTQ+ individuals by state and region in the United States. This would be done in an effort to further gain evidence and establish an evidence-base for L2B-Q specifically. To make the future directions clearer in our manuscript, we have added the following on page 14, lines 575-579: “Further pilot testing of L2B-Q utilizing randomized-controlled methods among rural LGBTQ+ adolescents, and among LGBTQ+ adolescents living in geopolitically conservative areas where stressors may be more intense and frequent, is a necessary next-step in helping mitigate health disparities among this population.”
Comment 5: [Lesbian/ gay/ .... among them who are having more stress? have you compared? if not can you compare and add a table?]
Response 5: The research team appreciates your suggestion at investigating between group differences in stress and outcome variables. We agree that investigating these differences between different sexual orientations and gender identities is paramount. However, comparisons of identity groups are beyond the scope of this particular paper. In addition, these analyses would fall beyond the statistical power for the sample size. It is our goal as a research team to continue researching the L2B-Q program with a larger sample size and perform a true RCT pilot trial, and it will be a top priority to not only investigate pre-post outcomes, but to also investigate differences in stress levels and differences in outcome changes by identity.
Thank you again, Reviewer 3, for all of your comments and feedback. We truly believe that your comments have made our manuscript stronger.